# Effects of Humic Acid Added to Controlled-Release Fertilizer on Summer Maize Yield, Nitrogen Use Efficiency and Greenhouse Gas Emission

Yanqing Guo [†], Zhentao Ma [†], Baizhao Ren, Bin Zhao, Peng Liu and Jiwang Zhang *

State Key Laboratory of Crop Biology, Agronomy College of Shandong Agricultural University, Taian 271018, China; 2019010046@sdau.edu.cn (Y.G.); 2019110156@sdau.edu.cn (Z.M.); bzren@sdau.edu.cn (B.R.); zhaobin@sdau.edu.cn (B.Z.); liup@sdau.edu.cn (P.L.)
* Correspondence: jwzhang@sdau.edu.cn; Tel./Fax: +86-538-8241-485
† These authors contributed equally to this work.

**Abstract:** Humic acid plays an important role in improving grain yield and reducing N losses. In order to explore the effects of humic acid added to controlled-release fertilizer on summer maize yield, nitrogen use efficiency (NUE) and the characteristics of greenhouse gas (GHG) emissions in maize farmland soil, a two-year field experiment was set up. The treatments consisted of two fertilizers: 3% humic acid added to controlled-release fertilizer (HACRF), controlled-release fertilizer (CRF) and a control (without N fertilizer, N0). The results demonstrate that the yield and NUE of summer maize were significantly increased with the addition of humic acid in N fertilizer. Compared with N0 and CRF, the yield of maize was increased by 99.1% and 5.0%, respectively. Compared with CRF, the contents of soil ammonium–nitrogen ($NH_4^+$-N) and nitrate–nitrogen ($NO_3^-$-N) in HACRF were higher during early maize growth stage but lower during the late grain-filling stage. The NUE and soil nitrogen interdependent rate of HACRF were significantly increased by 4.6–5.4% and 7.2–11.6%, respectively, across the years compared with those of CRF. Moreover, the annual cumulative $N_2O$ emissions in HACRF were decreased by 29.0% compared with the CRF. Thus, the global warming potential and greenhouse emission intensity of HACRF were significantly decreased by 29.1% and 32.59%, respectively, compared with CRF. In conclusion, adding humic acid to controlled-release fertilizer can result in higher yield and nitrogen uptake, improve nitrogen use efficiency and reduce greenhouse gas emissions, which have better yield and environmental effects.

**Keywords:** summer maize; humic acid; yield; nitrogen use efficiency; greenhouse gas emissions; fertilization

## 1. Introduction

Excessive emissions of greenhouse gases aggravate global warming, which is increasingly triggering a serious global environmental problem. Carbon dioxide ($CO_2$), methane ($CH_4$) and nitrous oxide ($N_2O$) are the main greenhouse gases (GHG), which contribute nearly 80% of gases to the greenhouse effect [1]. Soils can be both sources and sinks of $CO_2$, $CH_4$ and $N_2O$ [2]. Total emissions from food systems may account for 25–30% of total GHG emissions [3]. Soil GHGs mainly result from root respiration, microbial respiration and soil fauna respiration, which can be influenced by fertilization, cultivation and irrigation in agricultural production activities [4–7]. How to reduce the environmental cost under the premise of ensuring stable and even increased grain yield has aroused widespread attention. The optimal management of nitrogen fertilizer, such as adjusting nitrogen fertilizer types, recycling organic wastes, adding soil conditioners and optimizing water and fertilizer management, is an important measure at present in agricultural production research.

Controlled-release fertilizers (CRF) have a more controlled rate of release based on the engineering of the coating type, which can better match plant N needs throughout the growing season, reduce N losses to the environment and thus increase N availability for

plants [8,9]. CRF can significantly reduce the total amount of greenhouse gas emissions relative to conventional fertilization [10,11]. Compared with conventional fertilization, CRF reduces $N_2O$ emission by 15.5% [12]. Yao et al. [13] found that CRF had no significant effect on soil $CH_4$ absorption compared to conventional fertilization. Banger et al. [14], using a meta-analysis, concluded that the application of N fertilizers in paddy soil increased $CH_4$ emissions, which can be reduced by slow/controlled-release fertilizer [12]. Combined application of organic and inorganic fertilizers mitigates GHG emissions [15,16], and biochar is widely used as a soil amendment to restrain greenhouse gas production in cropland soils [17]. Agegnehu et al. [18] pointed out that the addition of organic modifiers significantly improved the physical and chemical properties of soil and may help to mitigate greenhouse gas emissions in certain systems while increasing maize yield. Therefore, we hypothesized that the emission reduction ability of CRF can be further improved by adding organic matter to fertilizer.

Humic acid is a kind of macromolecular organic matter produced by aerobic fermentation of plant residues [19]. It has many aromatic structures, phenolic hydroxyl structures and carboxyl structures, which make humic acid faintly acid and show solubility, electrification, adsorbability, ion exchange and complexation chelating properties [20]. The addition of humic acid to fertilizer can significantly increase crop yield, improve soil physical and chemical properties, increase soil adsorption capacity for $NH_4^+$, promote microbial activity, increase soil organic carbon content and fix inorganic nitrogen into organic nitrogen [21,22]. After the application of urea with humic acid, the carboxyl group and phenolic hydroxyl group of humic acid interact with the amide group of urea to form a complex with high stability, which improves the availability of $NH_4^+$-N and $NO_3^-$-N in soil and nitrogen use efficiency and reduces nitrogen loss [23–26]. Compared with urea, urea with humic acid significantly reduces $N_2O$ emissions [27]. Yu et al. [28] pointed out that the option of chemical N fertilizer fully substituted by organic N fertilizer can reconcile low climatic impact and high nitrogen agronomic efficiency (NAE). However, some studies have pointed out that humic acid can increase soil organic matter, which is not only the main carbon source of soil respiration but also an important source of $CH_4$ and $N_2O$ production [29]. Therefore, humic acid fertilizers promote greenhouse gas emissions [15,30,31]. CRF is a good fertilizer for emission reduction and efficiency improvement; however, whether adding humic acid to CRF can further improve this ability remains to be studied. The aims of this study were, therefore, to test the hypotheses that adding humic acid to CRF (HACRF) (1) improves summer maize yield, (2) promotes nitrogen uptake and utilization and (3) mitigates greenhouse gas emissions.

## 2. Materials and Methods

### 2.1. Experimental Site and Design

This study was conducted in 2019 and 2020 at the test demonstration base of Heyuan Seed Technology Co., Ltd. in Mazhuang, Tai'an, and the State Key Laboratory of Crop Biology, China (117°09′ E, 36°20′ N). The site has a temperate continental monsoon climate with an annual average temperature of approximately 16.4 °C and accumulated precipitation of 601.3 mm, which occurs mainly from June to August. The meteorological conditions during the summer maize growth stage are shown in Figure 1. The meteorological data were obtained through the field meteorological real-time monitoring platform of Jixing farm Wuxin station in Tai'an, Shandong Province. The meteorological station was set in our experimental field. The location is characterized by brown loam, where the basic nutrient content of organic matter was 15.18 g·kg$^{-1}$, total nitrogen was 1.86 g·kg$^{-1}$, $NO_3^-$-N was 10.00 mg·kg$^{-1}$, $NH_4^+$-N was 2.19 mg·kg$^{-1}$ and pH was 6.8 at the 0–20 cm soil layer. The experiment was arranged as a randomized block design with three replicates, each of which was 72 m$^2$, and it included the following three treatments: humic acid adding to 210 kg N/ha controlled-release fertilizer (HACRF), in which the content of humic acid was 3%; 210 kg N/ha controlled-release fertilizer (CRF); and a control (without N fertilizer, N0). The experiment material was Denghai 618 (mid-early hybrid), and the planting density

was 75,000 plants·ha$^{-1}$, which were sowed on June 15 and harvested on October 3. In the growth season of summer maize, N, $P_2O_5$ and $K_2O$ were applied at concentrations of 210, 52.5 and 67.5 kg·ha$^{-1}$, respectively. Our fertilizer was a finished product provided by Shandong Agricultural University Fertilizer Sci. & Tech. Co., Ltd. in Tai'an, China. The ratio of N to $P_2O_5$ to $K_2O$ in CRF was 28:7:9. The ratio of N to $P_2O_5$ to $K_2O$ in N0 was 0:7:9. The fertilizers were applied once during sowing. The nitrogen fertilizer type was controlled-release fertilizer, which regulates nitrogen release rate through coating technology, and the control ratio was 30%, and the nitrogen release time was 60–70 days. During the growing period of maize, good field management was practiced according to the production habits of local farmers, and micro spray belt was used for irrigation.

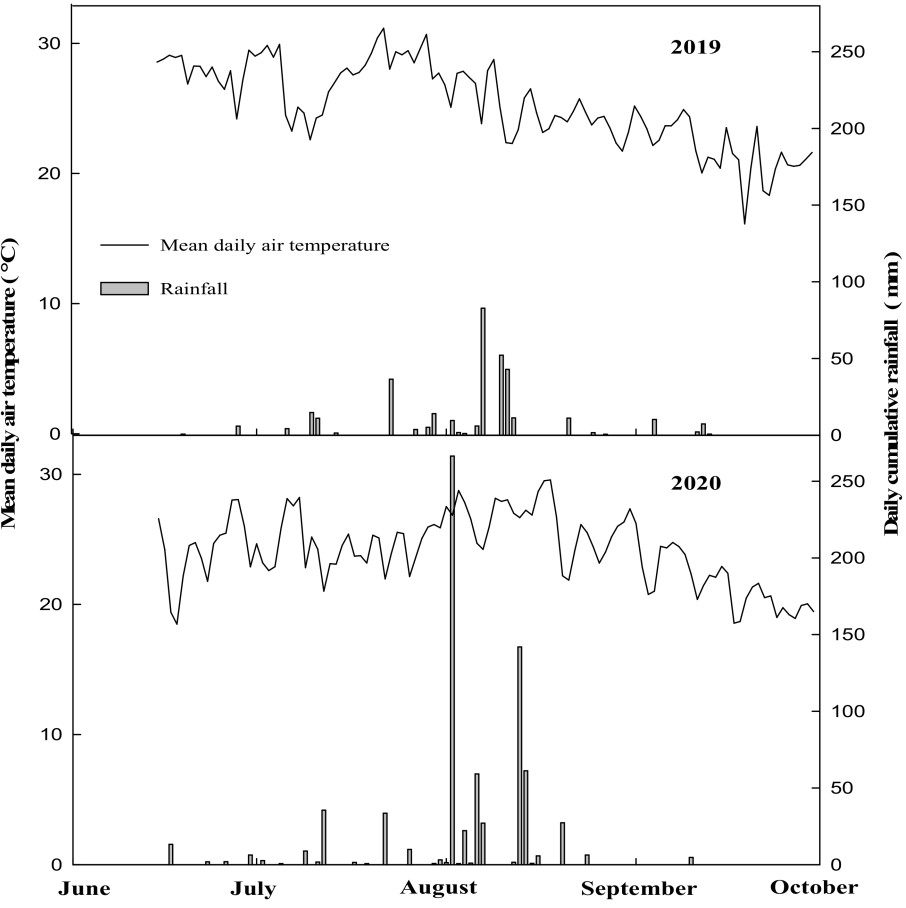

**Figure 1.** Meteorological conditions during the summer maize growth stage in 2019 and 2020.

*2.2. Measurements*

2.2.1. Grain Yield and Production Value

To determine maize yield and ear traits, 30 ears were harvested at the physiological maturity stage (R6) from three rows at the center of each plot. All kernels were air-dried, and grain yield was measured at 14% moisture, the standard moisture content of maize in storage or for sale in China (GB/T 29890-2013). According to the local market price (N0 input value, 2183 USD/ha; CRF input value, 2485 USD/ha; and HACRF input value, 2519 USD/ha; corn in 2019 was 0.25 USD/kg, 0.30 USD/kg in 2020 and 0.40 USD/kg in 2021),the production value, economic benefits and the ratio of production to investment were calculated.

2.2.2. GHG Emission

$N_2O$, $CH_4$ and $CO_2$ were collected using static chamber–gas chromatographic techniques. Two chambers were set for each treatment, which were placed between the rows

of corn. The chamber was insulated using sponge material and aluminum foil, which was enclosed by plastic sheets, and an air vent was installed in the middle of chamber. A pedestal was placed under chamber, which was sealed with water to ensure that the external environment did not affect the interior of chamber during the air extraction process. The dimensions of the chamber were $0.40 \times 0.40 \times 0.2$ (length $\times$ width $\times$ height). Gas samples (50 mL) were collected once every two days within a week after fertilization, and then once a week until corn harvest for collection at 9:00–11:00 a.m. using glass syringes from chamber headspace 0, 10, 20 and 30 min after soil sample was covered. The concentrations of $N_2O$, $CH_4$ and $CO_2$ in the collected samples were detected by an Agilent GC7890 (Agilent, California, USA) gas chromatograph [32].

The $N_2O$, $CH_4$ and $CO_2$ fluxes were calculated according to the linear variation in $N_2O$, $CH_4$ and $CO_2$ concentrations during the sampling process, and the formula for calculation is as follows [33]:

$$F = \rho \times H \times dC/dT \times 273/(273 + T)$$

where F is gas emissions flux ($mg \cdot m^{-2} \cdot h^{-1}$), $\rho$ is gas density in standard state ($g \cdot cm^{-3}$), H is the height of headspace in chamber (m), $dC/dT$ is the increment of gas concentration in unit time ($ppm \cdot h^{-1}$) and T is the absolute temperature during sampling.

The calculation formula of accumulated greenhouse gas emissions is as follows [33]:

$$CF = \sum_{i=1}^{n} \left( \frac{Fi + 1 + Fi}{2} \right) \times (t_{i+1} - t_i) \times 24$$

where CF is the accumulated greenhouse gas emissions, the unit of $N_2O$ is kg $N \cdot ha^{-1}$, $CH_4$ and $CO_2$ are given in kg $C \cdot ha^{-1}$, *F* is the first measurement of gas emission flux, unit is the same as above, 24 is the conversion coefficient from hourly emissions to daily emissions, $t_{i+1} - t_i$ is the number of days between two consecutive measurements and n is the total number of observations.

The calculation formula of global warming potential (GWP) is as follows [28]:

$$GWP = CF_{N2O} \times 265 + CF_{CH4} \times 28$$

where GWP is the global warming potential (kg $CO_{2\text{-eq}} \cdot ha^{-1}$) and $CF_{N2O}$ and $CF_{CH4}$ are the accumulated emissions of $N_2O$ and $CH_4$, respectively. $CH_4$ and $N_2O$ are considered as 28- and 265-fold, respectively, the GWP of $CO_2$ for 100 years [34,35].

The calculation formula of greenhouse gas intensity (GHGI) is as follows [28]:

$$GHGI = GWP/Y$$

where GHGI is global warming potential per unit output (kg $CO_{2\text{-eq}} \cdot kg^{-1}$) and Y is crop yield (kg $\cdot ha^{-1}$).

### 2.2.3. Nitrogen Accumulation and Utilization

At maturity stage (R6), the five representative plant samples were randomly selected from each treated plot and divided into stems and leaves. Samples were placed in the oven at 105 °C until green and then dried at 80 °C to constant weight. Nitrogen content of samples was measured by AA3 continuous flow analyzer (SFA CFA FIA BRAN+LUEBBE III) (AutoAnalyzer 3, SEAL Analytical GmbH, Norderstedt, Germany). Nitrogen accumulation and utilization were calculated as follows [36,37]:

Total N accumulation amount (TNAA, kg$\cdot$ha$^{-1}$) = plant nitrogen content amount $\times$ plant biomass

N partial factor productivity (NPFP, kg$\cdot$kg$^{-1}$) = grain yield/N rate

N use efficiency (NUE, %) = grain yield/(available nitrogen of soil in current season + N rate) $\times$ 100

$$\text{Soil nitrogen dependency rate (SNDR, \%)} = \text{(total N uptake by plant with applied N0/total N uptake by plant with applied N)} \times 100$$

### 2.2.4. $NH_4^+$-N and $NO_3^-$-N of Soil

Soil samples were extracted by using an earth drill from a 0 to 60 cm depth in each plot, which were divided into three layers with a height of 20 cm, to measure the concentrations of $NH_4^+$-N and $NO_3^-$-N at jointing stage (V6), booting stage (V12), tasseling stage (VT), milking stage (VT + 30) and maturity stage (R6). Soil was extracted with 1 M KCl and filtered through a 0.45 μm membrane filter to remove insoluble particulates. The contents of soil $NH_4^+$-N and $NO_3^-$-N were measured by AA3 continuous flow analyzer. Before sowing, 0–20 cm of soil was extracted to measure the foundation fertility [38].

### 2.2.5. Statistical Analysis

Dates were analyzed by one-way analysis of variance (one-way ANOVA) procedure using SPSS 17.0 software (SPSS, Inc., Chicago, IL, USA) with $p \leq 0.05$ considered significant, and SigmaPlot 10.0 was used to make figures.

## 3. Results

### 3.1. Yield and Its Components

Grain yield of summer maize was significantly increased by applying nitrogen, which was further improved with the addition of humic acid in N fertilizer. The trend of yield change in 2020 was generally the same as that in 2019. With the addition of humic acid, the HACRF resulted in 4.8% and 5.1% higher yields than that of CRF in 2019 and 2020, respectively. The main reason for the increased yield of summer maize with nitrogen application was that it increased the thousand-kernel weight and kernel number per ear, while the higher grain yield with the addition of humic acid in CRF was due primarily to increased thousand-kernel weight. Compared with N0, the application of CRF significantly improved the economic benefits of summer maize. Moreover, the application of humic acid improved the production value of summer maize. The output value, economic benefits and the ratio of production to investment of HACRF treatment improved by 4.9%, 15.5% and 3.8% across the years, respectively, compared to those of CRF treatment (Table 1).

**Table 1.** Effects of humic acid added to controlled-release fertilizer on the yield and production values of summer maize.

| Year | Treatment | 1000-Grain Weight (g) | Grains per Ear | Ears (No·ha$^{-1}$) | Grain Yield (t·ha$^{-1}$) | Production Value (USD·ha-1) | Economic Benefits (USD·ha$^{-1}$) | The Ratio of Production to Investment (USD·ha$^{-1}$) |
|------|-----------|----------------------|----------------|---------------------|---------------------------|----------------------------|-----------------------------------|------------------------------------------------------|
| 2019 | N0 | 277.6 c | 319 b | 69,997 a | 6.2 c | 1559 c | −624 c | 0.71 c |
| | CRF | 328.2 b | 529 a | 72,989 a | 12.7 b | 3187 b | 701 b | 1.27 b |
| | HACRF | 336.5 a | 536 a | 73,645 a | 13.3 a | 3339 a | 820 a | 1.32 a |
| 2020 | N0 | 337.9 c | 291 b | 66,945 a | 6.6 c | 1986 c | −196 c | 0.90 c |
| | CRF | 392.4 b | 446 a | 65,834 a | 11.5 b | 3478 b | 993 b | 1.39 b |
| | HACRF | 399.1 a | 457 a | 66,389 a | 12.1 a | 3652 a | 1133 a | 1.44 a |
| | Year (Y) | ** | ** | * | * | ** | ** | ** |
| | Treatment (T) | ** | ** | NS | ** | ** | ** | ** |
| | Y × T | NS | NS | NS | * | * | * | ** |

N0: without N fertilizer; CRF: controlled-release fertilizer; HACRF: adding humic acid to controlled-release fertilizer. Different letters in each column indicate significant differences at $p < 0.05$ (LSD). * Significant at the 0.05 probability level. ** Significant at the 0.01 probability level. NS: not significant.

### 3.2. Nitrogen Accumulation and Utilization

The TNAA and NUE were significantly increased by the addition of humic acid in fertilizer. Compared with CRF, the total nitrogen accumulation in HACRF-treated soil was significantly increased by 7.7% and 11.4% in 2019 and 2020, respectively. Compared with CRF, the NUE and NPFP resulting from HACRF treatment were significantly increased by 4.6% and 5.3% in 2019 and by 4.6% and 5.4% in 2020, respectively. However, the SNDR of HACRF-treated soil was significantly decreased by 7.2% and 11.6% in 2019 and 2020, respectively, compared to that of CRF-treated soil (Table 2).

**Table 2.** Effects of humic acid added to controlled-release fertilizer on nitrogen accumulation and utilization of summer maize.

| Year | Treatments | TNAA (kg·ha$^{-1}$) | NPFP (kg·kg$^{-1}$) | SNDR (%) | NUE (%) |
|---|---|---|---|---|---|
| 2019 | N0 | 109.2 c | - | - | - |
| | HACRF | 258.1 a | 63.2 a | 42.3 b | 47.4 a |
| | CRF | 239.7 b | 60.4 b | 45.6 a | 45.3 b |
| 2020 | N0 | 91.3 c | | | |
| | HACRF | 225.6 a | 57.8 a | 40.5 b | 41.3 a |
| | CRF | 199.8 b | 54.9 b | 45.8 a | 39.2 b |
| Year (Y) | | ** | ** | NS | ** |
| Treatment (T) | | ** | ** | ** | ** |
| Y × T | | NS | NS | NS | NS |

N0: without N fertilizer; CRF: controlled-release fertilizer; HACRF: adding humic acid to controlled-release fertilizer; TNAA: total N accumulation amount; NPFP: N partial factor productivity; SNDR: soil nitrogen dependency rate; NUE: nitrogen use efficiency. Different letters in each column indicate significant differences at $p < 0.05$ (LSD). ** Significant at the 0.01 probability level. NS: not significant.

### 3.3. $NH_4^+$-N and $NO_3^-$-N Content of Soil

With the advance of the growing stages, the contents of $NH_4^+$-N and $NO_3^-$-N in soil were first increased and then decreased, and significantly increased in the early growth stage of maize. The content of $NH_4^+$-N in soil reached the maximum at the tasseling stage. Compared with the $NH_4^+$-N content of CRF-treated soil, HACRF-treated soil at 0–20 cm and 20–40 cm soil layers was significantly increased by 22.0% and 22.7% in 2019 and by 18.9% and 25.3% in 2020, respectively. The content of $NH_4^+$-N in the 40–60 cm soil layer of HACRF-treated soil was significantly increased by 25.9% in 2019, while there was no significant difference in 2020, compared to that of CRF-treated soil. At the maturity stage, there was no significant difference in soil $NH_4^+$-N content between HACRF- and CRF-treated soil (Figure 2). Compared with CRF-treated soil, the content of $NO_3^-$-N in 0–20 cm soil layer at the booting stage and jointing stage was significantly increased by HACRF, while the content of $NO_3^-$-N in the 0–20 cm soil layer was decreased after anthesis. At maturity, compared with CRF, the $NO_3^-$-N content in 0–20 cm soil layer of HACRF-treated soil was significantly decreased by 26.1% and 13.5% in 2019 and 2020, respectively (Figure 3).

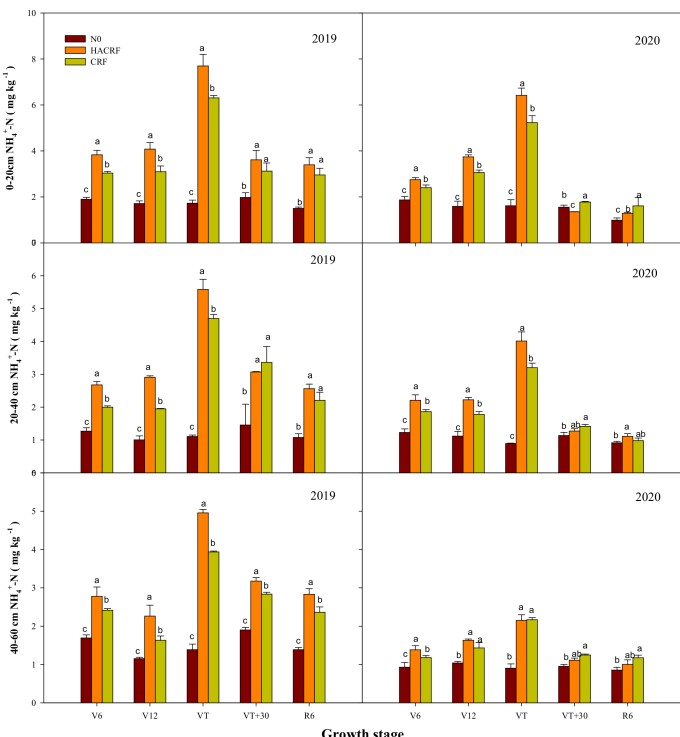

**Figure 2.** Effects of humic acid added to controlled-release fertilizer on $NH_4^+$-N during summer maize growth stage. N0: without N fertilizer; CRF: controlled-release fertilizer; HACRF: adding humic acid to controlled-release fertilizer. Different letters in each column indicate significant differences at $p < 0.05$ (LSD).

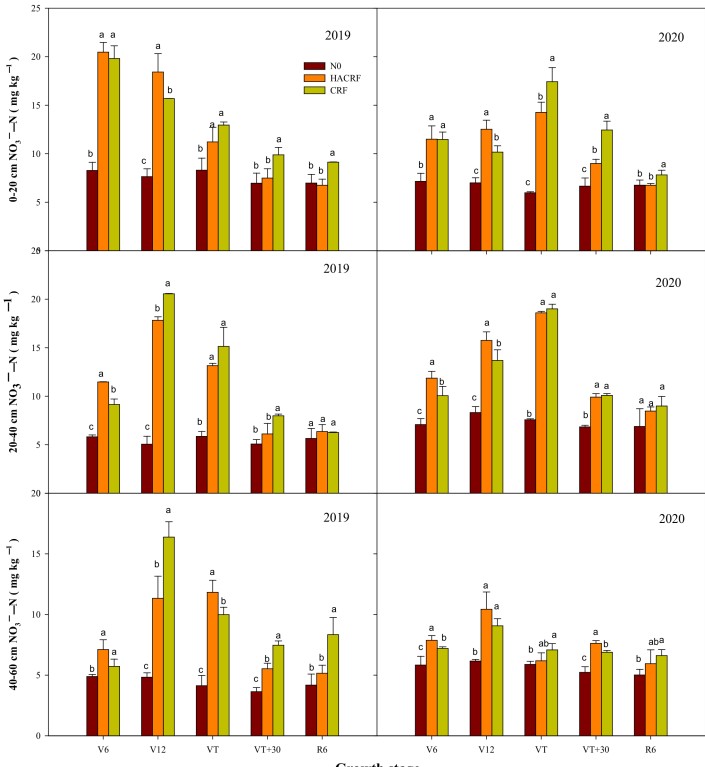

**Figure 3.** Effects of humic acid added to controlled-release fertilizer on $NO_3^-$-N during summer maize growth stage. N0: without N fertilizer; CRF: controlled-release fertilizer; HACRF: adding humic acid to controlled-release fertilizer. Different letters in each column indicate significant differences at $p < 0.05$ (LSD).

### 3.4. GHG Emission

3.4.1. Dynamic Change in Gas Emission Flux

During the whole maize season, there was no obvious $N_2O$ emission peak from N0 treatment, and the emission was very low. The soil $N_2O$ emission peak was significantly increased by nitrogen application, which was significantly reduced with the addition of humic acid to CRF (Figure 4). The peaks of $N_2O$ emission flux from CRF and HACRF treatments appeared on the 28th and 46th days after fertilization in 2019. Compared with the peak values of CRF, those of HACRF were significantly reduced by 24.5% and 10.9%, respectively. The peak of $N_2O$ emission flux from CRF treatments appeared on the 30th and 49th days after fertilization, while the peak of $N_2O$ emission flux from HACRF treatments appeared on the 28th and 49th days after fertilization in 2020. Compared with the peak values of CRF, those of HACRF were significantly reduced by 44.9% and 32.1%, respectively.

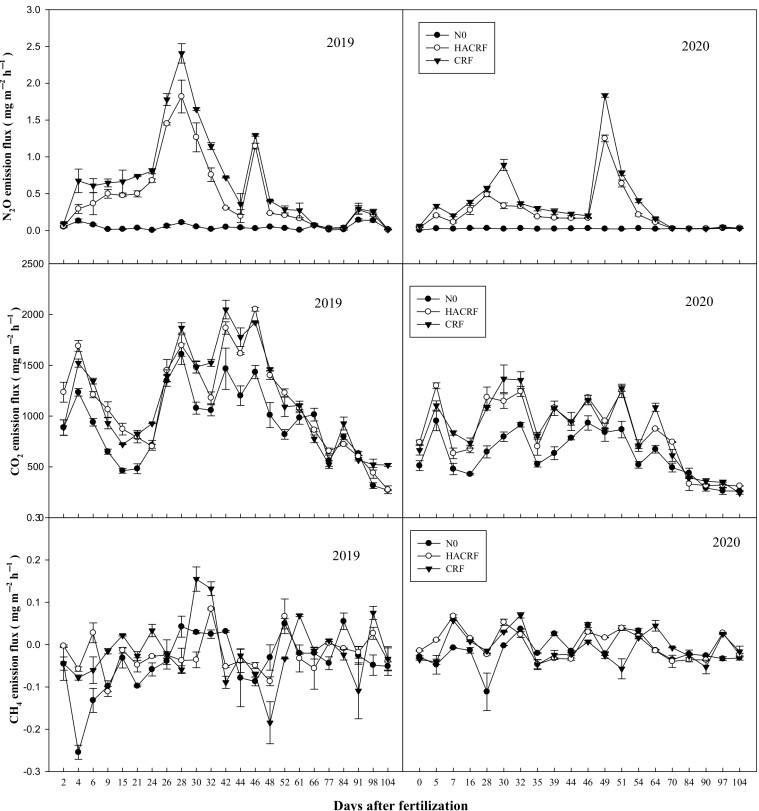

**Figure 4.** Effects of humic acid added to controlled-release fertilizer on greenhouse gas emission fluxes during summer maize growth in 2019 and 2020. N0: without N fertilizer; CRF: controlled-release fertilizer; HACRF: adding humic acid to controlled-release fertilizer.

In the whole maize season, soil $CO_2$ emission showed the same tendency in different treatments, as follows: increased firstly, then decreased, and then increased and finally decreased. The soil $CO_2$ emission was mainly concentrated in the 26–64 days after fertilization, and the first peak of $CO_2$ emission flux appeared on the 4th and 5th days after fertilization, which was significantly higher for HACRF than for CRF. To a certain extent, the soil $CO_2$ emission was increased by nitrogen application, which was significantly reduced with the addition of humic acid to CRF in most periods, and the change trend in 2020 was generally the same as that in 2019 (Figure 4).

In the whole maize season, soil $CH_4$ emission showed a fluctuating change, with positive and negative fluxes after fertilization. In general, the emission of $CH_4$ showed an absorption mode, which was promoted by nitrogen application to a certain extent compared with no nitrogen application, and the change trend in 2020 was generally the same as that in 2019 (Figure 4).

### 3.4.2. Cumulative Gas Emissions and Global Warming Potential

The total GHG emissions were significantly increased by nitrogen application, which was significantly decreased with the addition of humic acid, and the change trend was the same in the two years. Compared with CRF, the cumulative emissions of $N_2O$ and $CO_2$ for HACRF treatment were significantly decreased by 29.0% and 2.4% in 2019 and by 28.8% and 2.9% in 2020, respectively. The total absorption of CH4 was increased by 193.3% and 60.0% in 2019 and 2020, respectively. The GWP and GHGI were significantly increased by nitrogen application, which was decreased with the addition of humic acid to fertilizer. Compared with CRF treatment, the GWP and GHGI of HACRF treatment were significantly decreased by 29.3% and 32.7% in 2019 and by 28.9% and 32.5% in 2020, respectively (Table 3).

**Table 3.** Effects of humic acid added to controlled-release fertilizer on the accumulated greenhouse gas emissions, GWP and GHGI.

| Years | Treatments | Total Cumulative Emission of $N_2O$ (kg N·ha$^{-1}$) | Total Cumulative Emission of $CO_2$ (kg C·ha$^{-1}$) | Total Cumulative Emission of $CH_4$ (kg C·ha$^{-1}$) | GWP (kg $CO_{2\text{-eq}}$·ha$^{-1}$) | GHGI (kg $CO_{2\text{-eq}}$·kg$^{-1}$) |
|---|---|---|---|---|---|---|
| 2019 | N0 | 1.16 c | 21,010 c | −0.82 c | 293 c | 0.047 c |
| | HACRF | 8.72 b | 25,580 b | −0.44 b | 2299 b | 0.173 b |
| | CRF | 12.29 a | 26,213 a | −0.15 a | 3253 a | 0.257 a |
| 2020 | N0 | 0.67 c | 14,872 c | −0.47 c | 165 c | 0.025 c |
| | HACRF | 4.95 b | 20,242 b | −0.16 b | 1308 b | 0.108 b |
| | CRF | 6.95 a | 20,838 a | −0.10 a | 1840 a | 0.160 a |
| | Year (Y) | ** | ** | ** | ** | ** |
| | Treatment(T) | ** | ** | ** | ** | ** |
| | Y × T | ** | ** | ** | ** | ** |

N0: without N fertilizer; CRF: controlled-release fertilizer; HACRF: adding humic acid to controlled-release fertilizer; GWP: global warming potential; GHGI: greenhouse emission intensity. Different letters in each column indicate significant differences at $p < 0.05$ (LSD). ** Significant at the 0.01 probability level. NS: not significant.

### 4. Discussion

Crop productivity can be improved by nitrogen application, but the N losses and environmental pollution risk will increase as the accumulated mineral N exceeds crops demand [39]. Improving the absorption and utilization rate of urea nitrogen in fertilizer plays a very important role in agricultural production and environmental protection [27,40,41]. Controlled-release fertilizer can regulate nutrient release synchronously with crop nutrient absorption, which can significantly improve the nitrogen absorption of maize compared with common urea [42]. Studies at home and abroad have shown that partial replacement of inorganic N with organic materials could further increase nitrogen accumulation [43,44]. Humic acid is an economically available organic macromolecular matter that can improve soil nutrients, stimulate plant growth, regulate plant metabolism and promote the absorption of nutrients by plants [45–51]. We found that adding humic acid to CRF can significantly improve the total nitrogen accumulation of summer maize. In addition, humic acids have a large specific surface area, complex surface structure and numerous functional groups, and therefore possess strong adsorbability, hydrophilicity and complexation chelating properties and are faintly acid, which can improve soil physical and chemical properties, enhance the ability of soil to retain nutrient ions, promote mineral nutrient absorption and improve the fertilizer utilization efficiency [52,53]. Suntari et al. [25] showed that the contents of $NH_4^+$-N and $NO_3^-$-N in soil 28 and 42 days after rice planting increased due to the addition of humic acid in urea. Chen et al. [54] found that humic acid urea fertilizer significantly increased nitrogen absorption and NUE compared with N treatment alone. In our study, the contents of $NH_4^+$-N and $NO_3^-$-N in soil at the stage of VT may be due to: (1) the nitrogen fertilizer type was controlled-release fertilizer, which regulates the release period of fertilizer by coating technology to meet the demand for nitrogen fertilizer in the key period of crops [8,9]; (2) the maize changed from vegetative growth to reproductive

growth at VT, and in that time the total absorption area and active absorption area of the root system reached its maximum [55]. Additionally, the content of $NH_4^+$-N in soil at the early stage was significantly increased, while the content of $NO_3^-$-N in the soil after the booting stage was reduced by HACRF compared to CRF. Thus, the NPFE and NUE of summer maize were significantly improved, and the SNDR was significantly reduced with the addition of humic acid to CRF. Nitrogen (N) is one of primary essential nutrient elements for maize growth and yield formation. The yield of summer maize significantly increased with the nitrogen application. The improvement of nitrogen availability and utilization efficiency can optimize the growth environment of maize and further improve crop yield. In our study, the yield of summer maize was significantly increased by about 5.0% with the addition of humic acid to CRF.

GHG emissions are affected by many factors, such as precipitation, soil temperature and humidity and organic matter content. Appropriate irrigation can significantly reduce $N_2O$ and $CO_2$ emissions [56,57]. $CH_4$ is produced in an anaerobic environment, and its absorption has a significant negative correlation with soil moisture content [58]. In our study, the precipitation in 2020 was significantly higher than that in 2019, resulting in lower $N_2O$ and $CO_2$ emissions and $CH_4$ uptake. However, in 2020, the moisture was concentrated in August, especially in mid- and early August, which saw continuous heavy rainfall that coincided with the flowering and pollination period of maize. Then, the seed setting and nutrients transport were seriously affected, which in turn reduced the yield. Nitrate is very mobile in soil and can be lost through the flow of water [59], resulting in serious groundwater pollution and reduced NUE. Humic acid is an economically available organic macromolecular matter that can improve soil physical and chemical properties. The nitrogen is transformed into $NH_4^+$-N and $NO_3$-N, which can be directly absorbed and utilized by plants through decomposition and mineralization after it is applied to the soil, providing sufficient nitrogen sources for nitrification and denitrification, and ultimately promoting the increase in $N_2O$ emissions [60,61]. Our results show that the $N_2O$ emission was significantly increased with nitrogen application, which was significantly reduced by about 29% with the addition of humic acid. The reason for this result may be that humic acid can promote the expression of genes related to nitrate absorption and assimilation in roots, reducing the pH on the surface of plasma membrane of root cells, so as to promote the absorption of $NO_3^-$ by roots. At the same time, it can also fix $NH_4^+$-N in fertilizer through abiotic action [50]. Thus, humic acid delays the release of urea by inhibiting nitrification and ammoniation of urea in soil, so as to reduce the $N_2O$ emission [27]. Compared with $N_2O$ and $CH_4$, $CO_2$ has a greater impact on the greenhouse effect. We found that the soil $CO_2$ emission was significantly reduced in most periods with the addition of humic acid to CRF, which may be because humic acid fertilizer can improve the composition and binding form of humus and store carbon in soil [62]. There was an obvious $CO_2$ emission peak that was appeared on the 4th and 5th day after fertilization. The $CO_2$ emission flux of HACRF was significantly higher than that of CRF, which may be related to humic acid increasing the content of organic matter in soil and improving soil microbial activity [18,63]. The emission of $CH_4$ was promoted by nitrogen application [14,64], which can be significantly decreased by the combined application of organic manure and inorganic fertilizer, as the organic fertilizer can increase soil total organic carbon, which can reduce $CH_4$ emission by changing the microbial community [18,65–69]. In our study, the emission of $CH_4$ showed an absorption mode and the absorption was promoted by the addition of humic acid.

As a simple relative index based on radiation characteristics, GWP is often used to estimate the potential effects of different GHGs on the climate system. In the estimation, $CO_2$ is regarded as a reference gas, and the increase or decrease in $N_2O$ and $CH_4$ emissions is converted into the $CO_2$ equivalent using GWP values [70]. In our study, the emissions of $N_2O$ were significantly decreased, while $CH_4$ absorption was promoted by HACRF compared to CRF; thus, the GWP was significantly reduced. Meanwhile, the yield of summer maize was significantly increased with the addition of humic acid, so the GHGI was significantly reduced.

## 5. Conclusions

In this experiment, the yield and economic benefits of summer maize were significantly improved by basal application of 210 kg/ha controlled-release fertilizer provided by Shandong Agricultural University Fertilizer Sci. & Tech. Co., Ltd. Adding 3% humic acid to CRF was conducive to increasing nitrogen accumulation, improving the use efficiency of nitrogen and reducing greenhouse gas emissions, which resulted in a better yield, economic benefits and environmental performance. It is suitable for large-scale promotion and use in the Huang Huai Hai region.

**Author Contributions:** Conceptualization and methodology, J.Z.; investigation, data curation and writing—original draft, Y.G.; investigation, Z.M.; project administration, B.R.; methodology, P.L. and B.Z. All authors have read and agreed to the published version of the manuscript.

**Funding:** This research was funded by the China Agriculture Research System of MOF and MARA(CARS-02-21), Shandong Central Guiding the Local Science and Technology Development (YDZX20203700002548), and Shandong Agricultural Application Technology Innovation Project (SD2019ZZ013).

**Institutional Review Board Statement:** Not applicable.

**Informed Consent Statement:** Not applicable.

**Data Availability Statement:** Not applicable.

**Conflicts of Interest:** The authors declare that they have no known competing financial interests or personal relationships that could have appeared to influence the work reported in this paper.

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
