# Peer review of "Effects of Humic Acid Added to Controlled-Release Fertilizer on Summer Maize Yield, Nitrogen Use Efficiency and Greenhouse Gas Emission"

_agriculture, doi:10.3390/agriculture12040448_

Round 1

Reviewer 1 Report

The manuscript explore the effects of humic acid on summer maize yield, nitrogen use efficiency (NUE) and the characteristics of greenhouse gas (GHG) emission in  maize farmland soil. The treatments consisted of two fertilizers: humic acid added to controlled release fertilizer (HACRF), controlled release fertilizer (CRF), and a control (without N fertilizer, N0). The experiment was concluded that by adding humic acid to controlled release fertilizer can obtain higher yield and nitrogen uptake, improve nitrogen use efficiency and reduce green-house gas emissions, which have better yield and environmental effects. While reviewing the manuscript, following point arises and is as:

  • Page 1, line 34,Elaborate GHG and subsequently use GHG instead of green house gases in entire manuscript.
  • Page 2, Line 72, Elaborate NAE
  • MM: What is the recommended dose of NPK in maize crop and what type of urea (either coated/uncoated) being used by farmers in that region. Line 111- Placed between the rows of corn plants.
  • Metrological data reveal that Rainfall during 2019 was less compared to 2020. Kindly correlate the result with rainfall data with yield, NUE and GHG emission in discussion.
  • Kindly insert the reference for each formula line 124, 129, line 137, line 142, line 150, Line 168 and so on.
  • Line 154, its Nitrogen use efficiency (NUE) instead of N utilization efficiency (NUtE). Kindly go through review of Rumesh Ranjan and Rajbir Yadav, 2019 – A review on NUE and cite on the manuscript also.
  • Result:
  • Table 1: There were no significant difference in Kernel No., 1000 kernel wt., Ears no. and Grain yield under CRF and HACRF. What might be the reason.
  • Also calculate benefit cost ratio of CRF and HACRF. Is there any economic feasibility for farmers in term of yield by adding humic acid.
  • Figure 2 and 3: Why VT stage shows high NO3 and NH4 uptake. Kindly comment scientifically in discussion section.
  • Figure 4: Why emission of CO2, N2O and CH4 were less in 2020 compared 2019. Comment.
  • Reference of Malyan et al., 2016, Review on GHG and humic acid is missing. Kindly insert.

Reviewer 2 Report

Interesting research results. Important for science and agricultural practice. When describing the results, pay attention to the interactions between the factor and the years. Often the differences that exist are only between the means for the factor. Write the text as requested by the Agriculture journal. The methods and methodology need to be supplemented with missing information. I have included detailed comments in the original text. Explain what scale of plant development (development stages) you used. The list of literature must be adapted to the requirements of the journal. Fill in the gaps in the literature, e.g. pages, DOI, etc. 

Reviewer 3 Report

I do find this work interesting and valuable. The manuscript is well written, however below are some suggestion for improving.

Line 91 -  should be rather „pH was 6.8”

Lines 96 – 100 – could you here explain in a more detailed way this fertilizer aplication? In which form P2O5 and K2O were added? You wrote that there were three treatments: humic acid adding to controlled release fertilizer (HACRF); controlled release fertilizer (CRF); and a control (without N fertilizer, N0)”. Does this mean that in control were only P2O5 and K2O added? How this humic acid was added exactly? Could you write more about this controlled release fertilizer also? Was this ready to use product? If yes, who was the manufacturer?

Table 1 - are these letters  (used for statistical differences) used separtely for each year, or both years were compared together? If  this is the first option (which is correct), where is „a” in 2020 for 1000-kernel weight and for 2019 kernels number per ear (if there is „ab” should be also „a”)? If this is the second option – something is wrong with these letters: how is this possible that for example 277.55 has letter „c” and 337,86 has also „c” while 328.20 (which is value between these two values) has „b”? Please check and correct. The same goes for other parameters. You have to clearly write below the table what is compared using this letters.

Line 188 – there are no values for NUE in N0 (Table 2), so you can not write that it was  significantly increased by applying nitrogen

Line 210 – „while there was no significant difference was found in 2020” – should be  f.ex. „while there was no significant difference in 2020”.

Line 213 – that is not true for V12 and 20 – 40 cm according to Figure 3 – please be more specific.

Lines 235 – 236  „Soil N2O emission was significantly increased by nitrogen application, which was significantly reduced as the addition of humic acid to CRF (Fig. 4)”. How do you know that humic acid reduced N2O emission significantly? – there isn’t any statistical analysis of results in Figure 4. Could you perform statistical analysis for it or optionally use another words for describing the differences during dynamics description. I mean „significantly” implies usually that there are statistically proven differences, which here are not, because there is no statistical analysis. The same goes for CO2 and CH4 description (Figure 4).
